# Endoscopic In Vivo Hyperspectral Imaging for Head and Neck Tumor Surgeries Using a Medically Approved CE-Certified Camera with Rapid Visualization During Surgery

**DOI:** 10.3390/cancers16223785

**Published:** 2024-11-10

**Authors:** Ayman Bali, Thomas Bitter, Marcela Mafra, Jonas Ballmaier, Mussab Kouka, Gerlind Schneider, Anna Mühlig, Nadja Ziller, Theresa Werner, Ferdinand von Eggeling, Orlando Guntinas-Lichius, David Pertzborn

**Affiliations:** 1Clinical Biophotonics & MALDI Imaging, Department of Otorhinolaryngology, Jena University Hospital, 07747 Jena, Germany; ayman.bali@med.uni-jena.de (A.B.); marcela.mafra@med.uni-jena.de (M.M.); anna.muehlig@med.uni-jena.de (A.M.); nadja.ziller@med.uni-jena.de (N.Z.); theresa.werner@med.uni-jena.de (T.W.); ferdinand.von_eggeling@med.uni-jena.de (F.v.E.); orlando.guntinas@med.uni-jena.de (O.G.-L.); 2Department of Otorhinolaryngology, Jena University Hospital, 07747 Jena, Germany; thomas.bitter@med.uni-jena.de (T.B.); jonas.ballmaier@med.uni-jena.de (J.B.); mussab.kouka@med.uni-jena.de (M.K.); gerlind.schneider@med.uni-jena.de (G.S.); 3Comprehensive Cancer Center Central Germany, 07747 Jena, Germany

**Keywords:** head and neck cancer, hyperspectral imaging, intraoperative, surgery, visualization

## Abstract

During head and neck cancer surgery, accurately identifying the borders of malignant tumors is crucial but remains challenging. This study investigates the use of hyperspectral imaging (HSI) to support surgeons in real-time tumor visualization during surgery. The research focuses on integrating HSI into the surgical workflow, emphasizing the optimization of this process to ensure efficiency and ease of use for medical staff. By facilitating faster and more accurate evaluation of tumor margins, this approach has the potential to enhance surgical precision, minimize incomplete tumor resections, and ultimately improve patient outcomes.

## 1. Introduction

Head and neck cancer ranks as the seventh most common cancer globally, according to GLOBOCAN 2020, with incidence rates on the rise [1,2]. Despite advancements in diagnostic methods and treatments, the overall five-year survival rate remains relatively low at approximately 50% [3]. Treatment typically involves a multidisciplinary approach, including surgery, radiotherapy, chemotherapy, immunotherapy, or combinations thereof. In the context of surgical resection for head and neck cancer, achieving negative tumor margins—defined as margins exceeding 5 mm—and accurately assessing these margins are crucial for improving patient outcomes. Margins that do not meet this 5 mm threshold are associated with a threefold increase in the relative risk of mortality compared to cases with negative margins [4,5,6]. The current standard for tumor margin assessment relies on the surgeon’s subjective evaluation, biopsies taken for frozen sections during surgery, and final histopathology. However, frozen sections have limited accuracy, are subjective, suffer from selection bias, and add additional time and cost to the procedure [7,8,9,10].

Given the critical need to improve surgical margin detection in head and neck cancer surgery, various techniques for both in vivo and ex vivo assessment are under investigation. Proposed solutions to enhance diagnostic precision range from spectroscopic approaches, such as Raman spectroscopy, to molecular techniques like mass spectrometry [11,12,13]. Parallel research efforts are also exploring the utility of optical coherence tomography, narrow-band imaging, ultrasound, and magnetic resonance tomography [14,15,16]. Hyperspectral imaging (HSI), widely used in fields such as remote sensing, mineral exploration, and environmental geology [17], is emerging as a promising tool in medical imaging due to its speed, non-invasiveness, non-ionizing nature, and label-free capability [18]. HSI facilitates the differentiation of materials based on their unique interactions with light, a property tied to their molecular composition, which has led to extensive research in the life sciences [19]. Such studies frequently use HSI to identify different organs and tissue types and to assess tissue perfusion in transplantation contexts [19,20].

The properties of HSI make it an ideal candidate for enhancing tumor diagnosis, particularly in areas suitable for direct optical or endoscopic inspection, such as skin cancer, gastrointestinal cancer, and head and neck cancer [18,21,22,23,24,25]. One of the initial applications of HSI in head and neck cancer surgery aimed to detect tongue tumors [26]. Early research by Halicek et al. [27] focused on identifying head and neck cancer in gross ex vivo specimens by analyzing hyperspectral images using deep convolutional neural networks, which were trained on samples from 20 patients. This group later expanded their approach, applying it to a larger cohort, identifying cancer in resected specimens from 82 patients with salivary gland or thyroid cancer [28] and 102 patients with squamous cell carcinoma (SCC) [29]. Eggert et al. also combined HSI with deep learning for tissue classification, based on in vivo measurements of the laryngeal, hypopharyngeal, and oropharyngeal mucosa in 98 patients [30]. Other methods include assessing tumor margins on both stained and unstained sections [31,32].

In many of these applications, the non-intuitive data acquired by HSI are analyzed using diverse machine learning approaches, ranging from classical machine learning to more modern, computationally demanding, data-intensive deep learning methods, often in the form of convolutional neural networks [33]. Since HSI is most commonly used in remote sensing, many algorithms were initially developed for this purpose and later adapted for medical imaging [34]. A recurring challenge in developing data processing tools for hyperspectral imaging (HSI) is the lack of large-scale datasets, which limits the effectiveness of deep learning approaches. However, once training datasets reach an adequate size, deep learning methods typically outperform classical techniques [23,34].

In the referenced studies, most hyperspectral imaging systems are categorized either as custom-built prototypes or developed in-house [23,27] or as commercial imaging systems lacking CE certification [25,31]. In this study, we investigated the in vivo application of HSI to enhance margin visualization during head and neck tumor surgery. We combined a CE-certified, commercially available HSI system with rapid data visualization technologies, integrating real-time medical insights and the expertise of the operating surgeon.

## 2. Materials and Methods

### 2.1. Patients

In this study, 15 patients undergoing curative head and neck tumor resection at the Department of Otorhinolaryngology, Jena University Hospital, Germany, received additional in vivo hyperspectral imaging of the tumor site. The study was conducted between September 2023 and April 2024, with each patient providing written informed consent. Inclusion criteria were age ≥18 years, (suspected) primary head and neck cancer, and written informed consent. Patients were excluded if they did not provide written informed consent. Histopathological assessments including TNM classification and P16 status were performed for all patients, with the predominant diagnosis being squamous cell carcinoma (n = 12). Biopsies from the remaining patients (n = 3) showed no evidence of cancer during histopathological analysis, leading to their exclusion. An overview of all patient characteristics is provided in Table 1.

### 2.2. The Hyperspectral Imaging System

HSI was performed using the CE-marked TIVITA™ Mini Imaging System (Diaspective Vision GmbH, Am Salzhaff, Germany), which is medically approved and classified as a Class I medical device according to MDR regulations, paired with a rigid 0° endoscope designed specifically for HSI applications (Karl Storz SE Co. KG, Tuttlingen, Germany). The hyperspectral camera captures images with a spatial resolution of 540 by 720 pixels. The measured spectrum ranges from 500 to 1000 nm with 100 distinct channels, resulting in a spectral resolution of 5 nm. Illumination is provided by light-emitting diodes integrated within the HSI system, and the image acquisition time per image is approximately 6 s.

The imaging system is operated through a dedicated computer integrated with the device, which stores hyperspectral data and enables data processing as described in subsequent sections. The recommended working distance for this system is 7.5 cm; however, we conducted measurements at distances ranging from 2 cm to 8 cm to optimize the balance between the technical capabilities of the imaging system and intraoperative constraints.

### 2.3. Hyperspectral Imaging During Surgery

Before surgery, suitable patients are identified based on the specific study criteria, and informed written consent is obtained. The operating room staff is notified in advance, ensuring that all necessary equipment, including the surgical arm, HSI endoscope, and a mouth gag, is arranged. On the day of surgery, the equipment setup is completed prior to patient anesthesia. Once anesthesia is administered, the devices are positioned and prepared for use.

To assess the impact of camera movement artifacts, the system was tested under three different conditions. First, it was used in handheld mode. Second, it was mounted on a fully mechanical surgical arm, which stabilizes the imaging system during measurements while allowing full range of motion for positioning. Third, an alternative setup employed a motorized surgical arm designed to lock the endoscope in place during imaging. The imaging setup, including the mechanical arm, is illustrated in the intraoperative setting in Figure 1.

Using the RGB live view from the imaging system, the surgeon selected the field of view. Initially, a single HSI was captured to assess lighting conditions based on the RGB image reconstructed from the HSI. If the image appeared significantly too dark or too bright, adjustments were made to either the light intensity or the working distance. Once optimal settings (field of view, light intensity, and working distance) were established, one or multiple HSIs were acquired. After completing the imaging step, the tumor was resected according to standard practice. The surgeon performing the resection was blinded to the HSI results.

Figure 2 provides a detailed visualization of the protocol stages described in the text, illustrating the three main phases: (A) the patient selection and consent phase, (B) the preparation process, and (C) the intraoperative HSI measurements.

### 2.4. Data Processing

Prior to any further processing, the raw hyperspectral reflectance measurements are automatically calibrated against the black and white standards of the cameras sensor as follows:$$I_{c}=\frac{I_{R}-I_{B}}{I_{W}-I_{R}}$$
with $I_{C}$ being the calibrated intensity, $I_{R}$ the measured reflected intensity, $I_{B}$ the measured black standard, and $I_{W}$ the measured white standard. These calibrated spectra are the basis for all further evaluation.

Each HSI measurement consists of 388,800 spectra, arranged to form a hyperspectral image with 540 by 740 pixels and processed through one of two different data processing pipelines. In the first pipeline, the HSI measurement was analyzed and displayed directly using the software provided by the HSI system vendor. This process generated four distinct views of the HSI: an RGB image reconstructed from the HSI data, and three false-color images representing tissue oxygenation (StO_2_), near-infrared perfusion index (NIR-PI), and tissue water index (TWI). The definitions of these parameters are detailed in ref. [35].

In the second data processing pipeline, the calibrated spectra were processed directly using an in-house-developed Python application. First, each individual spectrum was normalized using standard normal variate (SNV) to eliminate multiplicative interferences from scatter, particle size variations, and uneven illumination [36]. The SNV transformation is calculated as follows:$$X_{T}=\frac{X-\bar{X}}{STD(X)}$$
with $X_{T}$ being the transformed spectrum, X the original spectrum, $\bar{X}$ the mean of X, and STD(X) the standard deviation of X.

Subsequently, unsupervised dimensionality reduction was conducted using either principal component analysis (PCA) or t-distributed Stochastic Neighbor Embedding (t-SNE) [37,38]. PCA applies orthogonal transformations to project correlated variables onto a set of orthogonal principal components, effectively reducing dimensionality while retaining the majority of data variance. This approach is commonly used to improve data interpretability and decrease computational requirements in large datasets [39].

On the other hand, t-SNE is a machine learning algorithm developed for visualizing high-dimensional data by reducing them to a low-dimensional space (two or three dimensions) suitable for display in a scatter plot. It is particularly effective for visualizing high-dimensional datasets and aids in identifying patterns and clusters.

False-color images were created based on PCA and t-SNE and these false-color images were combined with the reconstructed RGB images. After this step, the resulting false-color images were divided into seven regions using k-means clustering. K-means clustering is a method of vector quantization, which aims to partition n observations into k clusters in which each observation belongs to the cluster with the nearest mean, serving as a prototype of the cluster.

An exemplary HSI hypercube stored as multichannel.tif file and RGB images representing different wavelength ranges as false-color images are provided as Appendix A. The HSI hypercube contains all 100 wavelength images in ascending order, where each can be viewed separately. The false-color images show the tissue oxygenation (StO_2_), near-infrared perfusion index (NIR-PI), and tissue hemoglobin index (THI), as described in ref. [35]. The wavelength ranges used to calculate these indices are given in the filenames.

### 2.5. Surgical Annotations

To evaluate the effectiveness of the second data processing pipeline, the overlays produced on the RGB images were compared to manual annotations made by an experienced surgeon. To ensure the independence of these annotations from the hyperspectral imaging (HSI) analyses conducted during the initial surgery, each RGB image was presented to a second surgeon, who was not involved in the original procedure (see Figure 3). This surgeon was tasked with marking areas as they would in a surgical context, categorizing them as ‘Tumor’ or ‘Non-Tumor’. These findings were subsequently cross-referenced with the histopathological assessment results of the tumor biopsies and resected specimens.

### 2.6. Evaluation

To compare the manual annotations of the surgeon with the unsupervised clustering results, the resulting clusters were transformed into a binary ‘tumor/non-tumor’ image. This process relies on expert knowledge: while experienced surgeons can easily identify the ‘main tumor area’, defining exact tumor margins remains a significant challenge. For this study, we instructed the surgeon to intuitively and quickly draw a small rectangle in the center of the main tumor area on the RGB image. Alternatively the request to the surgeon was to point out spots from which to take a tumor biopsy. All clusters that shared at least one pixel with the marked area were designated as ‘tumor,’ while everything else was classified as ‘non-tumor’.

Next, the accuracy, sensitivity, and specificity of this approach were calculated for each measurement by comparing the classes derived from unsupervised clustering with the perceived tumor distribution on a pixel-by-pixel basis.

Additionally, we developed a pipeline allowing the surgeon to mark a region of the image for comparison with the rest of the image. For this marked area, we calculated the average spectrum and then computed the spectral similarity between this area and each other spectrum in the image. Spectral similarity was determined using two different metrics: the spectral angle mapper (SAM) [40] and the gradient-based spectral similarity measure (GSSM) [41]. This information was then converted into a false-color image, illustrating the similarity between the selected region and every other point in the hyperspectral image.

## 3. Results

### 3.1. In Vivo Hyperspectral Imaging During Head and Neck Tumor Resection

After the first two to three surgeries, during which the medical personnel familiarized themselves with handling the hyperspectral imaging system, the additional time required for hyperspectral imaging during surgery was reduced to under 5 min. This time encompassed setting up the surgical arm to hold the endoscope, preparing the HSI camera and endoscope, and completing the imaging process. The learning curve for the hyperspectral imaging system is illustrated in Figure 3, which displays the tumor detection accuracy achieved for each patient in chronological order.

Handling the endoscope and imaging system was straightforward for the participating surgeons. The availability of the RGB live view facilitated navigation with the endoscope, aiding in locating the resection site and selecting the field of view. One issue encountered in this process was a slight discrepancy in the field of view between the RGB imaging system and the HSI system, occasionally resulting in images capturing only part of the relevant area. The evaluation of working distances indicated that minor deviations from the recommended 7.5 cm distance—up to approximately 3 cm—had minimal impact on imaging quality. At the recommended working distance, the measured field of view was 86 mm by 66 mm.

The imaging process itself took approximately six seconds. While this duration allowed for reasonably good handheld imaging with minimal motion artifacts, the surgeons preferred a system that locked the endoscope’s position during imaging, which resulted in overall higher image quality. Of the two holding arms, the mechanical arm was favored by the surgeons due to its ability to provide more precise positioning of the endoscope. Since the procedure closely resembled a standard endoscopic investigation, no complications were anticipated or recorded. Excluding the additional setup time, the entire hyperspectral imaging procedure was completed in approximately 2–3 min per patient, including selecting the imaging site, positioning the endoscope, and performing the imaging itself.

### 3.2. Analysis of the HSI and Comparison with Surgical Annotations

We evaluated the data processing by comparing it with the annotations performed by a surgeon and by assessing the suitability of the developed data processing pipeline for intraoperative application.

While the standard parameters (oxygenation (StO_2_), near-infrared perfusion index (NIR-PI), and tissue water index (TWI)) were automatically calculated and recorded, none showed a strong correlation with the surgical annotations. The clustering approach, guided by the surgeon’s expertise, achieved an accuracy of 79%, a sensitivity of 72%, and a specificity of 84%, and visually aligned well with tumor localization, as shown in Figure 4. Between the two clustering methods compared, PCA and t-SNE, the results were similar. Given that t-SNE requires substantial computational resources without offering clear benefits, it was omitted in later analyses, and all presented results are based on PCA.

Displaying the spectral similarities of a marked region to other parts of the hyperspectral image as false-color images produced results qualitatively comparable to those of the unsupervised clustering but did not yield additional insights. When processing the hyperspectral data cubes directly on the computer controlling the imaging system, the results of the unsupervised clustering could be overlaid on the white-light RGB image within 10 s after initiating the imaging process. The final visualization, as it appears intraoperatively, is shown in Figure 5.

## 4. Discussion

In vivo visualization of head and neck tumors remains one of the main challenges in surgical margin evaluation. In this study, we demonstrated that hyperspectral imaging (HSI) is a feasible tool for rapid in vivo visualization of tissue differences during head and neck tumor surgery. Our optimized protocol limits the additional surgery time to between two and three minutes and reduces the time from the start of imaging to the full visualization of tissue clustering to less than 10 s. Although the primary aim of this work was to establish the feasibility of this workflow, we achieved promising results in terms of accuracy and specificity for in vivo tumor detection (accuracy: 79%; sensitivity: 72%; specificity: 84%).

The comparison in Table 2 highlights the significant advantages of hyperspectral imaging (HSI) over other imaging modalities. HSI stands out for its label-free and non-invasive application, unlike fluorescence imaging, which requires contrast agents, and CLE, which is more invasive. Additionally, HSI provides exceptionally high spectral resolution, superior to NBI, and CLE, making it highly effective for detecting subtle tissue variations. While its spatial resolution is moderate, HSI’s combination of speed, spectral detail, and real-time feasibility makes HSI an excellent tool for intraoperative use, offering robust support in head and neck oncological diagnostics.

While comparable studies have shown that head and neck cancer can be detected using hyperspectral imaging [24,26,42], none of these previous studies conducted in vivo measurements using a CE-certified imaging device that meets the requirements for clinical investigational devices as outlined in the Medical Device Regulation. Moreover, these studies were primarily proofs of concept and did not fully address the specific challenges of the intraoperative environment, highlighting a significant gap in translating findings into clinical practice. In contrast, our study includes measurements from multiple regions—such as the oral cavity, oropharynx, larynx, and hypopharynx—enabling analysis across a broader range of tissue types and anatomical complexities. This wider scope improves the generalizability of our findings and provides strong validation for the use of hyperspectral imaging across the full spectrum of head and neck oncological conditions. Finally, prior studies often involve long measurement times, analyses limited to ex vivo samples, and complex computational methods [18]. Other studies using the same imaging system have mostly focused on perfusion parameters, limiting their applicability [43].

By focusing on establishing a repeatable, reliable, and fast workflow, we successfully reduced the additional surgery time to under three minutes and demonstrated that visual feedback on tissue distribution can be provided within ten seconds of initiating the imaging process. Additionally, we used the system’s tissue classification performance as an indicator of the surgeon’s familiarity with the imaging protocol. This allowed us to show that surgeons can become proficient with the system after only two surgeries. In comparison, a large-scale multicenter clinical trial utilizing fluorescence visualization for oral cancer margin localization recommends reviewing the imaging quality for the first 10 surgeries at each site, alongside hands-on training with at least three patients. Additionally, the fluorescence visualization protocol requires the presence of a fluorescence visualization expert on site, separate from the surgeon [44]. Beyond workflow advantages, hyperspectral imaging may also offer improved detection accuracy compared to autofluorescence imaging or fluorescence imaging using agents like 2-NBDG and proflavine [45].

In addition to HSI and fluorescence visualization, other approaches include narrow-band imaging (NBI) [46]. NBI is of particular interest due to its similarities and differences with HSI. Like HSI, NBI leverages the spectral characteristics of various tissues; however, it focuses on specific spectral bands rather than a broad spectrum. NBI uses only two wavelength bands (400–430 nm and 525–555 nm) to highlight vascular structures [47]. While a recent systematic review concluded that there is insufficient evidence to claim that NBI is more accurate than white-light-guided surgery for determining safe surgical mucosal margins, the understanding of specific, significant spectral bands could be valuable in further HSI analysis [46]. This knowledge could enable the selection of only a subset of available wavelengths, potentially achieving faster imaging.

Another alternative method is confocal laser endomicroscopy (CLE), which can be used with or without a contrast agent to visualize cellular structures and detect tissue differences [48]. While CLE integrates well with the surgical workflow, it is limited by a very small field of view, covering less than 1 mm^2^. Despite the high accuracy, the impact of CLE is currently limited by its small field of view [49].

At this stage of the study, several inaccuracies and challenges remain. One challenge arises from the presence of surgical instruments, which, due to their reflective surfaces, mirror surrounding tissue, leading to misclassification in the unsupervised tissue clustering. Additional inaccuracies stem from the surgical annotations themselves, which are prone to error and limited to the surface layer of the tissue, while the higher wavelengths captured by the hyperspectral imaging system contain information from deeper tissue layers. Although this limitation exists with current surface-only annotations, HSI’s penetration depth presents an opportunity to enhance tumor margin detection by incorporating information from these deeper layers—a capability successfully demonstrated ex vivo in resected breast cancer specimens [50]. Finally, while our data processing workflow incorporates real-time surgical expertise, even more domain knowledge could be leveraged by pretraining a neural network on a large-scale dataset, potentially capturing the collective knowledge of multiple domain experts. For such an approach to be effective, however, a significantly larger dataset would be required, and the hyperspectral image analysis would need to align with the time constraints of the intraoperative setting.

## 5. Conclusions

With the introduction of the first CE-certified endoscopic hyperspectral imaging system, large-scale trials are necessary to fully assess the potential of this technique, especially with more advanced data analysis approaches. Our study offers a baseline evaluation and highlights several important findings: the available system can be seamlessly integrated into the routine surgical workflow and, at least within our small patient cohort, provides meaningful results accessible in real time during head and neck tumor surgery. This integration is achieved without long wait times, specific lighting requirements, disruptive changes to the surgical workflow, costly equipment, complex computational methods, or extensive personnel training—a combination not currently available with any other imaging method.

Consequently, we see a clear path forward and a strong need for further work in this direction to be initiated. After more extensive data collection, this could lead to enhanced tumor visualization during head and neck cancer surgery and ultimately improved patient outcomes.

## Figures and Tables

**Figure 1 cancers-16-03785-f001:**
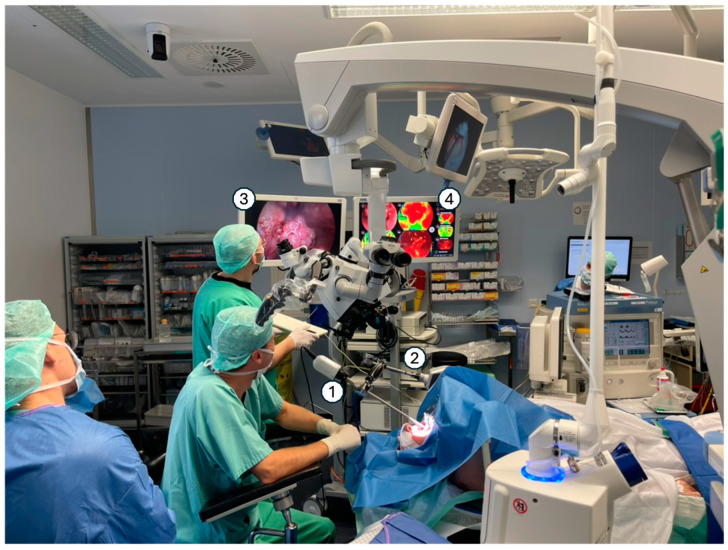
Hyperspectral imaging system setup for intraoperative use. Hyperspectral imaging system in the intraoperative setting including the following: 1: HSI-Camera with attached endoscope. 2: Motorized surgical holding arm. 3: Screen with endoscopic white light image. 4: Screen showing real-time generation of the HSI measurement.

**Figure 2 cancers-16-03785-f002:**
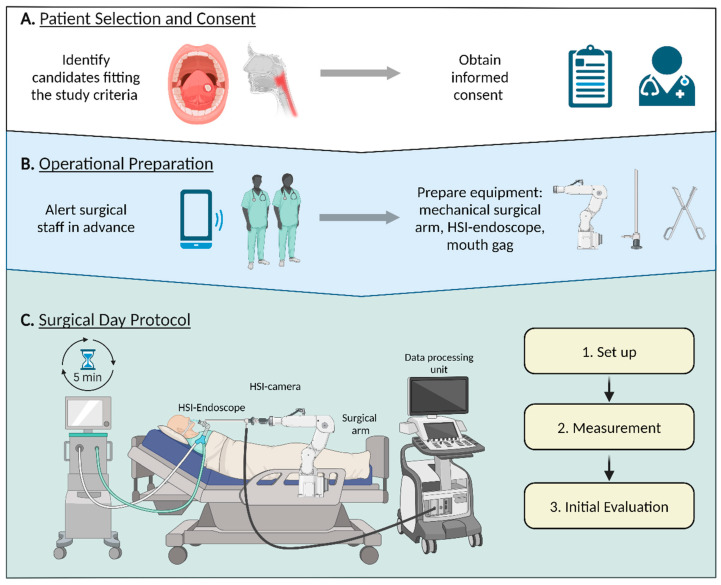
Integration of hyperspectral imaging in surgical procedures: workflow and protocols. Hyperspectral imaging surgical integration. (**A**): Patient selection and consent. (**B**): Operational preparation. (**C**): Surgical day protocol in the OR: 1. Set up and calibrate instruments prior to surgery, ensuring they are standby-ready following anesthesia administration. 2. Coordinate with the surgeon to determine optimal measurement sites post-anesthesia and perform measurements. 3. Evaluate the initial measurements for quality assurance, checking for adequate lighting.

**Figure 3 cancers-16-03785-f003:**
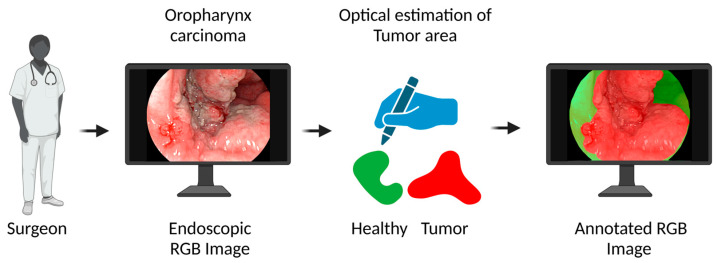
Surgeon’s tumor estimation on RGB endoscopic images. Independent surgeon’s optical estimation of tumor areas on RGB endoscopic images with tumor marked in red and non-tumor tissue in green.

**Figure 4 cancers-16-03785-f004:**
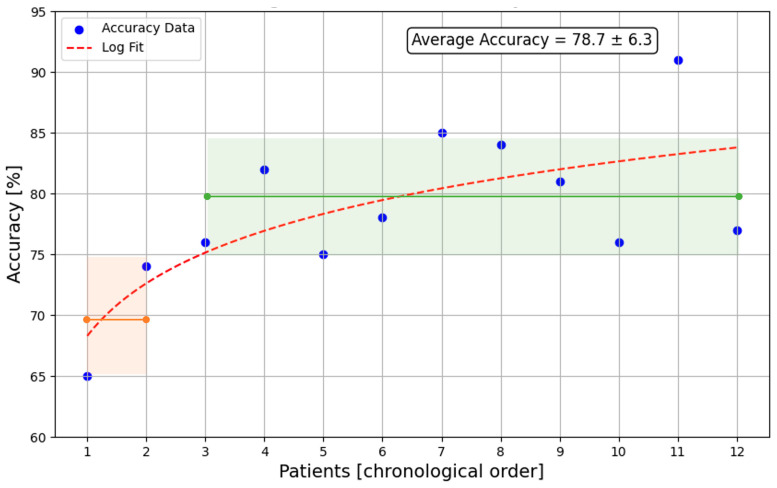
Improvement in HSI workflow accuracy with surgeon experience. Change in reported accuracy of the HSI workflow with the experience of the surgeon. The accuracy is calculated as described in the Materials and Methods section. Each point represents one patient. Dotted line: logarithmic fit of the data points. Orange line: average of first two patients. Green line: average of the next ten patients. Shaded area: standard deviation.

**Figure 5 cancers-16-03785-f005:**
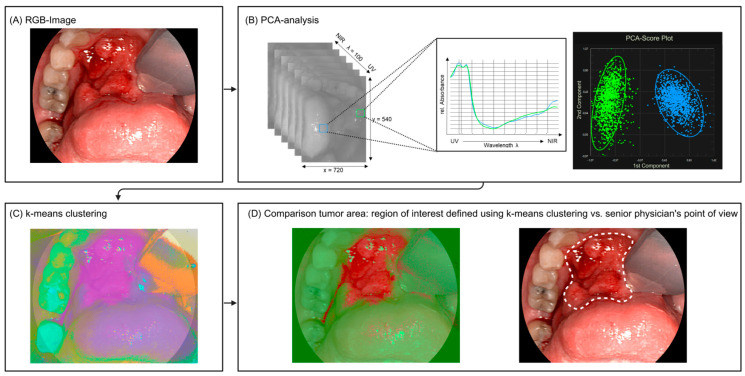
Hyperspectral imaging analysis and clustering of oral cavity squamous cell carcinoma. Hyperspectral imaging measurement of a cT4a cN2 cM0 oral cavity squamous cell carcinoma. (**A**) RGB image (a standard image showing the visual appearance of the tumor). (**B**) The spectral absorption curves of healthy (blue) and tumor (green) tissue are presented. These curves are plotted against the first principal component (PCA), effectively visualizing the separation of the two tissue classes based on their distinct spectral properties. (**C**) K-means clustering analysis (n = 7), performed using PCA data to differentiate various tissue regions. (**D**) Results of the unsupervised clustering based on region of interest (left) vs. senior physician’s annotations (right).

**Table 1 cancers-16-03785-t001:** Patient and histopathological characteristics (n.a.: not analyzed) of 12 patients. Patients where the final histopathological assessment ruled out squamous cell carcinoma were excluded and are not shown. Each patient’s TNM classification, p16 status, and final histopathological assessment are shown.

No	Localization	Right/Left	TNM	Gender	p16	Histopathology
1	Oropharynx	Left	pT2 pN M0	Male	p16+	Squamous cell carcinoma
2	Larynx	Right	pT3 pN3b cM0	Female	n.a.	Squamous cell carcinoma
3	Oropharynx	Right	pT4 cN2 cM0	Male	p16+	Squamous cell carcinoma
4	Larynx	Right	pT3 pN2b cM0	Male	p16-	Squamous cell carcinoma
5	Oropharynx	Left	pT2 pN1 cM0	Male	p16+	Squamous cell carcinoma
6	Oral cavity	Right	pT2 cN0 cM1	Male	n.a.	Squamous cell carcinoma
7	Larynx	Right	cT1b cN0 cM1	Male	p16-	Squamous cell carcinoma
8	Larynx	Right	pT2 N0 cM0	Male	n.a.	Squamous cell carcinoma
9	Oral cavity	Right	pT3 pN3b M0	Male	p16-	Squamous cell carcinoma
10	Oropharynx	Left	pT3 pN3b cM0	Male	p16-	Squamous cell carcinoma
11	Oropharynx	Left	cT3 cN2b cM0	Male	p16+	Squamous cell carcinoma
12	Oral cavity	Right	cT4a cN2 cM0	Male	p16-	Squamous cell carcinoma

**Table 2 cancers-16-03785-t002:** Comparison of imaging modalities used in clinical practice, including hyperspectral imaging (HSI), narrow-band imaging (NBI), fluorescence imaging, and confocal laser endomicroscopy (CLE). The table compares these methods based on various criteria, considering the shortest possible imaging time suitable for the surgical environment, including factors such as label-free capability, in vivo feasibility, spatial and spectral resolution, and imaging speed.

**Criterion**	**HSI**	**NBI**	**Fluorescence Imaging**	**CLE**	
**Label-Free**	Yes	Yes	No	No	
**In Vivo Feasibility**	Yes	Yes	Yes (depends on contrast agent)	Yes (limited to small areas)	
**Spatial Resolution**	Medium *	High	High	High	
**Spectral Resolution**	Very High	Low	Medium	Low	
**Speed**	High	High	High	Medium	
**Imaging modality**	**Label-Free**	**In Vivo Feasibility**	**Spatial Resolution**	**Spectral Resolution**	**Speed**
**HSI**	Yes	Yes	Medium*	Very High	High
**NBI**	Yes	Yes	High	Low	High
**Fluorescence Imaging**	No	Yes (depends on contrast agent)	High	Medium	High
**CLE**	No	Yes (limited to small areas)	High	Low	Medium

* Prioritizes spectral detail, with a trade-off in spatial resolution.

## Data Availability

All presented data will be made available upon reasonable request.

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
