# Peer review of "Endoscopic In Vivo Hyperspectral Imaging for Head and Neck Tumor Surgeries Using a Medically Approved CE-Certified Camera with Rapid Visualization During Surgery"

_cancers, 2024, doi:10.3390/cancers16223785_

Round 1

Reviewer 1 Report

Comments and Suggestions for Authors

Excellent study, exactly what the field of HSI needs. This study is extremely important in advancing HSI as a viable clinical imaging modality.

Comments on the Quality of English Language

Please recheck English language.

Author Response

Comment 1: Excellent study, exactly what the field of HSI needs. This study is extremely important in advancing HSI as a viable clinical imaging modality.

Response 1: We thank the reviewer for this positive evaluation of our study. We hope that our work can play a part in bringing hyperspectral imaging into the clinical setting.

Response to Comments on the Quality of English Language

Point 1: Please recheck English language.

Response 1: We revised the manuscript with strong focus on the quality of English language. We thank the reviewer for pointing our attention in this direction since it has helped us to improve the overall quality of our manuscript.

Reviewer 2 Report

Comments and Suggestions for Authors

The authors have achieved the in-vivo visualization of malignant tumor margins using appropriate HSI instruments for rapid detection during surgery. While the detection results hold clinical significance, there are still some issues that need to be addressed.

1. The mentioned “spectral range is 500 nm” is ambiguous. Just mentioned the range is much clearer for understanding.

2. The manuscript mentioned that each HSI measurement consists of 388.800 spectra. It should be “388,800”. Meanwhile, these are more like pixels in different spatial. Indeed, considering the spectral dimension, each pixel can capture signals at different wavelengths, but current description is challenging to understand how to analyze them separately by pixel.

3. The manuscript presents RGB images. Are there any original hyperspectral images available that can be displayed across individual spectral bands?

4. The author mentions using PCA or t-SNE for dimensionality reduction. When exactly is PCA used, and in what situations is t-SNE applied? Is the time consuming only one difference between these two methods?

5. The use of the mechanical arm effectively reduced motion artifacts caused by the operator. However, in actual use, there are still deviations introduced by factors such as pulse and blood flow. How was this part addressed?

6. Whether it is PCA analysis or K-means clustering, these are algorithms that can only categorize data and cannot provide predictive labels. How were the mentioned evaluation parameters, such as accuracy, calculated?

7. In line 287, the manuscript mentioned “but did not provide further insights”. What specific mapping process was used?

8. Are there any other algorithms that were used in the discrimination of tumor area? The accuracy has clinical value, but there is still room for improvement.

9. In Fig.4, What specific significance does the logistic fitting curve have? It seems that there is no connection between each patient, making it unclear what information this curve represents.

Author Response

The authors have achieved the in-vivo visualization of malignant tumor margins using appropriate HSI instruments for rapid detection during surgery. While the detection results hold clinical significance, there are still some issues that need to be addressed.

We thank the reviewer for this assessment and the suggestions below, which have helped us improve the quality of the manuscript.

Comment 1: The mentioned “spectral range is 500 nm” is ambiguous. Just mentioned the range is much clearer for understanding.

Response 1: We clarified this part in the section 2.2. It now reads: The measured spectrum ranges from 500 to 1000 nm with 100 distinct channels resulting in a spectral resolution of 5 nm.

Comment 2: The manuscript mentioned that each HSI measurement consists of 388.800 spectra. It should be “388,800”. Meanwhile, these are more like pixels in different spatial. Indeed, considering the spectral dimension, each pixel can capture signals at different wavelengths, but current description is challenging to understand how to analyze them separately by pixel.

Response 2: We corrected “388.800” to “388,800” and added the following clarification in section 3.4: Each HSI measurement, consists of 388,800 spectra, arranged to form a hyperspectral image with 540 by 740 pixel and …

Comment 3: The manuscript presents RGB images. Are there any original hyperspectral images available that can be displayed across individual spectral bands?

Response 3: Due to the difficulties with displaying hyperspectral images we refrained from showing them directly in the paper. Nonetheless we agree with the reviewer that there is a need for showing original hyperspectral images and have included an exemplary full data cube and three RGB images, averaging over relevant spectral regions in the supplements.

Comment 4: The author mentions using PCA or t-SNE for dimensionality reduction. When exactly is PCA used, and in what situations is t-SNE applied? Is the time consuming only one difference between these two methods?.

Response 4: We clarified in the manuscript, that all presented results are based on PCA. The following part can now be found in the results section: Between the two clustering methods compared, PCA and t-SNE, results were similar. Given that t-SNE requires substantial computational resources without offering clear benefits, it was omitted in later analyses, and all presented results are based on PCA.

Comment 5: The use of the mechanical arm effectively reduced motion artifacts caused by the operator. However, in actual use, there are still deviations introduced by factors such as pulse and blood flow. How was this part addressed?

Response 5: We thank the reviewer for this insightful question. While pulse and bloodflow certainly influence the hyperspectral imaging process we did not observe a strong influence in our study. Our assumption is that the combination of macroscopic imaging and lowered heart rate during anesthesia reduced these effects to the point where they did not cause problems.

Comment 6: Whether it is PCA analysis or K-means clustering, these are algorithms that can only categorize data and cannot provide predictive labels. How were the mentioned evaluation parameters, such as accuracy, calculated?

Response 6: We improved and clarified this process in the first paragraph in section 2.6. In short, the expertise of an independent surgeon was used to identify the cluster(s) containing tumor by asking the surgeon to point at the middle of the main tumor area. The rationale behind this approach is the practical knowledge that only the identification of the tumor borders is challenging, not the general localization of the tumor.

Comment 7:  In line 287, the manuscript mentioned “but did not provide further insights”. What specific mapping process was used?

Response 7: We clarified this paragraph in section 3.2. It now reads as: Displaying the spectral similarities of a marked region to other parts of the hyperspectral image as false color images produced results qualitatively comparable to those of the unsupervised clustering but did not yield additional insights. 

Comment 8: Are there any other algorithms that were used in the discrimination of tumor area? The accuracy has clinical value, but there is still room for improvement.

Response 8: We did not yet implement additional algorithms for the discrimination of tumor areas but are planning to do so once more measurements are collected. Our assumption, as mentioned in the discussion, is that more data allows the implementation of deep learning based analysis which outperforms the currently implemented approach. 

Comment 9: In Fig.4, What specific significance does the logistic fitting curve have? It seems that there is no connection between each patient, making it unclear what information this curve represents.

Response 9: We use the logistic fitting curve as a simplified way to represent the learning curve of the operating surgeon with the hyperspectral imaging system. Logarithmic models are commonly used to model human learning. [1]

1 FE Ritter, LJ Schooler. The learning curve. International encyclopedia of the social and behavioral sciences. 2001;13:8602-5.

Reviewer 3 Report

Comments and Suggestions for Authors

1. Line 106: Mentioned that ≥18 years age. Any reason about perticular age ?

2. Did you tried more than 60 age.?

3. In table 1. P16 means?

4. The attempt of the work is impressive. Conclusion last paragraph not reliable. Better to remove and add one line sentence like."further extension work to be initiated..".

Could be accepted after minor concerns.

Comments on the Quality of English Language

Minor editing needed 

Author Response

Comment 1: Line 106: Mentioned that ≥18 years age. Any reason about perticular age ?

Response 1: Head and neck cancer is very rare in minors. Therefore, in this study we focused on adult patients.

Comment 2: Did you tried more than 60 age.?

Response 2: The study included patients above the age of 60 but the age of the patient was not used as additional information for the analysis. In a larger scale study we see the potential to include age as an additional factor for analysis, due to the assumption that age could have a measurable effect in the tissue spectra of the patient.

Comment 3:  In table 1. P16 means?

Response 3: The expression of P16 is commonly used as a surrogate marker for HPV infection in head and neck cancer and assessed during histopathological evaluation [2]. We clarified the mention of P16 status as part of the histopathological assessment in section 2.1.

Comment 4: The attempt of the work is impressive. Conclusion last paragraph not reliable. Better to remove and add one line sentence like."further extension work to be initiated..".

Response 4: We improved the last paragraph of the conclusion taking into account the reviewers suggestion. 

Response to Comments on the Quality of English Language

Response 1: We revised the manuscript with strong focus on the quality of English language. We thank the reviewer for pointing our attention in this direction since it has helped us to improve the overall quality of our manuscript.

Round 2

Reviewer 2 Report

Comments and Suggestions for Authors

The authors have addressed all my concerns and just one further suggestion. The mentioned HSI hypercube, which is stored as multichannel .tif file in the supplement material needs more description. For example, in the given image, how are the multi-channel or multi-spectral bands divided? Or is the displayed result a composite image of broad spectral bands? Since this Cuba dataset is three-dimensional, and the author presents it as a two-dimensional image, a more detailed data description is needed.

Author Response

Comment 1: The authors have addressed all my concerns and just one further suggestion. The mentioned HSI hypercube, which is stored as multichannel .tif file in the supplement material needs more description. For example, in the given image, how are the multi-channel or multi-spectral bands divided? Or is the displayed result a composite image of broad spectral bands? Since this Cuba dataset is three-dimensional, and the author presents it as a two-dimensional image, a more detailed data description is needed.

Response 1:

Again we thanks the reviewer for the valuable feedback took the reviewers suggestions to heart. We added further detail explaining supplement S1. The description now reads: 

An exemplary HSI hypercube stored as multichannel .tif file and RGB-images representing different wavelength ranges as false color images are provided as supplement S1. The HSI hypercube contains all 100 wavelength images in ascending order, which each can be viewed separately. The false color images show the tissue oxygenation (StO2), near-infrared perfusion index (NIR-PI), and tissue hemoglobin index (THI), as described in ref. (35). The wavelength ranges used to calculate these indices are given in the filenames.